# Surface Roughness Prediction of Titanium Alloy during Abrasive Belt Grinding Based on an Improved Radial Basis Function (RBF) Neural Network

**DOI:** 10.3390/ma16227224

**Published:** 2023-11-18

**Authors:** Kun Shan, Yashuang Zhang, Yingduo Lan, Kaimeng Jiang, Guijian Xiao, Benkai Li

**Affiliations:** 1AECC Shenyang Liming Aero-Engine Co., Ltd., No. 6 Dongta Street, Dadong District, Shenyang 110862, China; zys800801@163.com (Y.Z.); lydlyd621@126.com (Y.L.); 2College of Mechanical and Vehicle Engineering, Chongqing University, No. 174 Shazheng Street, Shapingba District, Chongqing 400444, China; jiangkmeng@outlook.com (K.J.); xiaoguijian@cqu.edu.cn (G.X.); 3School of Mechanical and Automotive Engineering, Qingdao University of Technology, No. 777 Jialingjiang Road, Huangdao District, Qingdao 266520, China; lbk17082@163.com

**Keywords:** titanium alloy, abrasive belt grinding, roughness prediction, neural network

## Abstract

Titanium alloys have become an indispensable material for all walks of life because of their excellent strength and corrosion resistance. However, grinding titanium alloy is exceedingly challenging due to its pronounced material characteristics. Therefore, it is crucial to create a theoretical roughness prediction model, serving to modify the machining parameters in real time. To forecast the surface roughness of titanium alloy grinding, an improved radial basis function neural network model based on particle swarm optimization combined with the grey wolf optimization method (GWO-PSO-RBF) was developed in this study. The results demonstrate that the improved neural network developed in this research outperforms the classical models in terms of all prediction parameters, with a model-fitting R^2^ value of 0.919.

## 1. Introduction

Titanium alloy is a very effective metal that has been utilized extensively in a variety of industries, including aerospace, vehicle production, medical equipment, and others [1,2,3]. However, due to their unique strength, hardness, and chemical stability, titanium alloys have consistently been difficult to machine. Among the various processing techniques, titanium alloy belt grinding has drawn a lot of attention as a popular and efficient processing technique [4]. Abrasive belt grinding realizes the flexible processing of titanium alloy workpieces, which achieves better surface quality and accuracy to meet the requirements of high-precision machining [5].

However, with the rapid expansion of manufacturing, processing, and other related industries, the quality standards for products are getting more and more strict, and the processing industry is increasingly shifting in favor of high efficiency, fine quality, and low cost [6,7,8]. To increase their competitiveness in the global market, manufacturers compete to produce “zero-defect” products, which necessitate parts with exceptional surface quality [9]. The modern processing industry will confront significant hurdles as a result of the difficulty in controlling surface roughness under various processing circumstances and the lack of clarity regarding the processing variables that influence surface roughness.

In recent years, with the advancement of intelligence, numerous sophisticated algorithms have been employed to forecast the surface roughness of workpieces and enhance their predictive power. The causes of high and low surface roughness, as well as roughness prediction and modeling, have currently been developed as a near-complete theoretical system [10,11,12]. Tian et al. [13] developed a prediction model for the association between different process factors and workpiece surface roughness using a BP neural network based on the experimental results. Li et al. [14] constructed a state parameter prediction model based on the BP neural network by performing grinding tests on samples of nickel-based superalloys, and the prediction accuracy of the model is 93.58%. Qi et al. [15] took the maximum cutting depth of the belt, the speed of the belt, and the feed rate of the workpiece as input parameters to establish the prediction model of polishing surface roughness of the belt based on the BP neural network. The results show that the predicted value is in good accordance with the experimental value. Different from the BP neural network, the RBF neural network [16,17,18] utilizes the Gaussian activation function, which can address some of the issues with the BP neural network, such as the lengthy training period, ease of local optimum, and so on. It can generalize well, make predictions quickly, and adapt better to various types of data.

However, researchers become dissatisfied with the direct use of classic algorithms when the study goes deeper and they discover flaws in the traditional algorithms. As a result, they start to think of ways to improve traditional algorithms. The most popular strategy is to combine optimization algorithms with conventional prediction models, and the most utilized particle swarm optimization (PSO) technique is the optimization algorithm. Zhang et al. [19] suggested a data-driven roughness prediction approach for the GH4169 superalloy and discovered that the PSO-BP-based roughness prediction algorithm had a positive impact on the prediction of the superalloy. Yang et al. [20] built the PSO-BP surface roughness prediction model by using the particle swarm optimization algorithm to optimize the initial weights and thresholds of the BP neural network. Wang et al. [21] designed a temperature prediction model using an RBF neural network and, then, used particle swarm optimization and Levenberg-Marquardt computation to create the PSO-LM-RBF prediction method with a reduced deviation of prediction results and a more stable model.

Therefore, to accurately predict the belt grinding surface roughness of titanium alloy, the PSO algorithm was utilized in this study to optimize the RBF network parameters, obtaining the optimal solution and improving the operation efficiency. In addition to this, the iterative formula was updated with the application of the GWO algorithm, which is employed to prevent the algorithm from losing its capacity to converge later on and enter the local optimum problem. Then, empirical programs and formulas are used to establish the structural framework of the neural network and to clarify the key parameter values that must be employed in the algorithm. Finally, the accuracy of the algorithm is verified by simulation.

The remainder of the paper is structured as follows: the second part describes the prediction method adopted in this study. The determination of experimental parameters and the experimental results are discussed and analyzed in the third part. Finally, we summarize the paper.

## 2. Methods

### 2.1. Data Acquisition

The experimental equipment was based on a precision CNC belt grinder (2MGY5580, SAMHIDA, Chongqing, China), as shown in Figure 1. The TC4 titanium alloy, with dimensions of 400 mm × 200 mm × 5 mm, and the #80 alumina belt were adopted for the experimental study. A single grinding length of 35 mm was used for the processing, which was conducted under constant pressure.

Researchers in this field have found that workpiece surface roughness in belt grinding is significantly influenced by variables including abrasive belt particle size, belt linear speed, feed speed, and grinding depth [22,23,24]. In particular, the belt particle size is significantly influenced by the test material and essentially stays the same throughout the test. Therefore, as stated in Table 1, three test factors are identified, including belt linear speed (*v*_s_), feed speed (*v*_p_), and grinding depth (*a*_p_), and four horizontal orthogonal tests are conducted.

The test scheme, as presented in Table 1, is designed, and a total of 64 grinding experiments are conducted in the pre-experiment. The selection of experimental parameters is mainly based on the performance of the machine tool and the maximum/minimum parameters based on engineering experience. In addition to this, the testing scheme takes into account the characteristics of the orthogonal test table and neural network (the more training samples, the higher the prediction accuracy). Therefore, we divided the parameters of vs. between 7.8–19.5 m/s, *v*_p_ between 200–500 mm/min, and *a*_p_ between 3–12 mm into four segments at equal intervals for orthogonal experiments. After grinding, a portable roughness measurement device was used to determine the surface roughness of the test titanium alloy. During the process, roughness detection is conducted for five different points, and the final surface roughness value is calculated by taking the arithmetic average of the five measurements. As a result, the dataset of grinding parameters and corresponding roughness was built. Table 2 displays some data from the dataset.

### 2.2. Data Pre-Processing

Normalization [25] of these collected data is required to eliminate units, balance orders of magnitude, and avoid sample features with low values from being unduly controlled. Test data in different units can affect the results. The normalization formula is as follows:(1)$$y=\frac{x_{i}-x_{\min}}{x_{\max}-x_{\min}}$$
where *x_i_* is the sample data collected, and *x*_max_ and *x*_min_ are the maximum and minimum values of the sample data collected.

### 2.3. Prediction Model Based on GWO-PSO-RBF

#### 2.3.1. RBF Neural Network

The RBF neural network uses a Gaussian excitation function, which can address some of the issues with classic BP neural networks, including their lengthy network training procedures and propensity to easily enter local optimums [17]. It has excellent generalization capabilities, and, for a given amount of input data, only a few neuron parameters and hidden layer weights are used in the operation, which significantly increases the prediction speed of RBF neural networks and renders them more flexible to diverse types of data.

There are three structural layers in the network used in this study. The first layer is used to input the machining parameters. In the second layer, the Gaussian function was used as the activation function to process the input parameters non-linearly. The third layer is the output layer to output the final roughness prediction results. This is shown in Figure 2.

The input vector is *x*(*t*) = (*x*_1_, *x*_2_, …, *x*_N_)*^T^* and there are three different types of processing parameters in this work, which is equal to the number of input vectors. The terms *x*_1_, *x*_2_, and *x*_3_ represent *v*_s_, *v*_p_, and *a*_p_, respectively. After the sample data pass through the intermediate layer, the output is a nonlinear activation function *h*_j_(*t*):(2)$$h_{j}(t)=\exp(-\frac{||x(t)-c_{j}(t)|{|}^{2}}{2\sigma_{j}^{2}}),j=1,2\cdots m$$
where *c_j_* is the center of the *j*th node, *σ_j_* is the width of the *j*th node, ||*x*(*t*) − *c_j_*(*t*)|| is the Euclidean distance between the sample and the node center, and m is the number of nodes in the middle layer. By weighting the output data of the intermediate layer, the roughness prediction result can be obtained, as shown in Equation (3):(3)$$y_{i}(t)=\sum_{j=i}^{m}\omega_{ji}h_{j}(t),i=1,2\cdots ,n$$
where *w* is the weight and *n* is the number of network outputs.

The parameters of the RBF neural network, such as *c_j_*, *σ_j_*, and *w*, need to be determined by iteration. To better optimize the above parameters, this study suggested a GWO-PSO hybrid optimization method to better optimize the parameters.

#### 2.3.2. Particle Swarm Optimization Algorithm

Three parameters in the RBF neural network need to be set artificially: weight, node center, and radial basis width. The traditional RBF neural network often uses gradient descent iterative optimization, but this method not only has a poor training effect, taking a long time, but, also, it easily falls into local optimum so that the global effect cannot reach the best position. The parameter selection of the RBF neural network is, essentially, an optimization process, so the PSO optimization algorithm is used to optimize the parameters that need to be set manually in the RBF neural network and select the optimal values.

The particle swarm optimization algorithm [26] is an optimization method, which optimizes parameters by limiting the process of birds foraging in nature, and, finally, realizes the “survival of the fittest”. The particle will continuously evolve during each update process, and the parameters it carries will adjust in line with this evolution. The particle will evolve into a new particle if the prediction error corresponding to the parameters of the evolved particle is lower than that of the non-evolved particle. On the other hand, the undeveloped particles are kept.

In this study, the position of the particle corresponds to the value of the parameters that the neural network needs to train. Particle fitness corresponds to the error size of the roughness prediction model. In the iterative process, each particle modifies its speed and direction to move closer to the parameter value that reduces the prediction error through guidance. The first is called individual extremum guidance and it refers to the parameter value that each particle in an iterative process determines to reduce its own mistake. The global extremum guidance is the parameter value that was attained by every particle during the iteration procedure with the minimum overall error.

These two extreme values are used by the particle swarm to determine how to update each particle’s parameter, and the update formula is as follows:(4)$$v_{ij}^{k+1}=\omega v_{ij}^{k}+c_{1}r_{1}(pbest_{ij}^{k}-x_{ij}^{k})+c_{2}r_{2}(gbest_{ij}^{k}-x_{ij}^{k})$$
(5)$$x_{ij}^{k+1}=x_{ij}^{k}+0.5\times v_{ij}^{k}$$
where $x_{i}^{k}=(x_{i1}^{k},x_{i2}^{k},\cdots ,x_{ij}^{k}),i=1,2,\cdots ,m$ is the parameter that the particle *i* contains, $v_{i}^{k}=(v_{i1}^{k},v_{i2}^{k},\cdots ,v_{ij}^{k})$ is the velocity of the particle *i*, *w* is the weight of the middle layer, *k* is the current iteration number, *c*_1_ and *c*_2_ are learning factors to balance the relative importance of two extreme values, and *r*_1_ and *r*_2_ are randomly assigned values between 0 and 1.
(6)$$\omega =\omega_{\max}-(\omega_{\max}-\omega_{\min})\times \frac{k}{k_{\max}}$$
where *k* represents the number of iterations so far and *k*_max_ represents the maximum number of iterations. The terms *w*_max_ and *w*_min_ represent the maximum and minimum weights, respectively, which are generally set to 0.9 and 0.4, respectively.

#### 2.3.3. Grey Wolf Encirclement Optimization Strategy

The particle swarm is easy to aggregate in the final iteration, which reduces its searchability and causes the roughness prediction result to fall into the local optimum. Additionally, depending solely on the global optimum to direct parameter iteration will cause the subsequent iterations to move more slowly and reduce the capacity for convergence. The grey wolf method [27] will be incorporated into this study to enhance the parameter iteration formula and address the issues.

The grey wolf algorithm will decide which three people in the group have the smallest prediction error during the iterative update. The problem of diminishing convergence performance in the later stages of the algorithm is improved since other particles will surround the three elite individuals rather than the single optimal individual. The bounding search strategy, which is the most significant search strategy in the grey wolf algorithm, has the following mathematical representation:(7)$$X(k+1)=X_{p}(k)-A\times D$$
(8)$$D=|C\times X_{p}(k)-X(k)|$$
where *t* is the number of iterations, *X_p_*(*t*) is the position vector of the prey, and *X*(*t*) is the position vector of the grey wolf. The schematic diagram of its surroundings is shown in Figure 3.

As shown in Figure 3, assuming that the grey wolf is located in (*X*, *Y*) and the prey is located in $\left(X^{\prime },Y^{\prime }\right)$, the grey wolf will move to $\left(X^{\prime }-X,Y^{\prime }\right)$ by Equations (7) and (8) when $\vec{A}=(1,0)$ and $\vec{C}=(1,1)$. Different values of the coefficients $\vec{A}$ and $\vec{C}$ will produce different bounding effects, as shown in Figure 3, where $\vec{A}$ and $\vec{C}$ are coefficient vectors, which can be expressed by Equations (9) and (10), as follows:(9)$$\vec{C}=2r_{1}$$
(10)$$\vec{A}=2ar_{2}-a$$
where *r*_1_ and *r*_2_ are random numbers of [0, 1], *a* is the control parameter linearly decreasing with the number of iterations in [0, 2], and the decreasing formula is as follows:(11)$$a=2(1-\frac{k}{k_{\max}})$$

The introduction above states that the grey wolf algorithm employs the elite group advice, choosing the best three elite individuals as follows: the best solution *α*, the second best solution *β*, and the third best solution *δ* to guide the particle parameters. The three elite individuals will guide the particles in the form of bounding according to Equations (9) and (10). The guiding strategy formula is as follows:(12)$$D_{\alpha }=|C_{1}\times X_{\alpha }(k)-X(k)|$$
(13)$$D_{\beta }=|C_{1}\times X_{\beta }(k)-X(k)|$$
(14)$$D_{\delta }=|C_{1}\times X_{\delta }(k)-X(k)|$$
where Equations (12)–(14) represent the distances between each particle and three elite individuals *α*, *β*, and *δ*, respectively; Equations (15)–(17) are the moving directions of particles to three elite individuals; and Equation (18) is used as the moving direction of particle swarm after combining the guidance of three elite individuals:(15)$$X_{1}(k)=X_{\alpha }(k)-A_{1}\times D_{\alpha }$$
(16)$$X_{2}(k)=X_{\beta }(k)-A_{2}\times D_{\beta }$$
(17)$$X_{3}(k)=X_{\delta }(k)-A_{3}\times D_{\delta }$$
(18)$$X(k+1)=\frac{X_{1}(k)+X_{2}(k)+X_{3}(k)}{3}$$

The performance of the particle swarm optimization algorithm can be enhanced by using the grey wolf algorithm bounding strategy in the update formula. Its updated formula will become Equation (20):(19)$$v_{ij}^{k}=\omega (X_{ij}^{k}-x_{ij}^{k})+c_{1}r_{1}(pbest_{ij}^{k}-x_{ij}^{k})+c_{2}r_{2}(gbest_{ij}^{k}-x_{ij}^{k})$$

According to Formula (19), the update strategy of GWO is incorporated into the position update formula of PSO based on maintaining individual experience and the group optimal guiding strategy of PSO, which can somewhat alleviate the issues with the update formula of PSO, the prediction process is shown in Figure 4.

## 3. Experimental Results and Discussion

### 3.1. Experiment Details

#### 3.1.1. Parameter Setting of RBF Neural Network

The nonlinear mapping issue in feedforward networks can be resolved by an intermediary layer, according to research and analysis [28,29,30]. As a result, there was just one intermediary layer in the RBF neural network used in this study. The number of nodes in the middle layer will influence the prediction effect of the network for different process parameters and prediction aims. This effect is, typically, calculated by the empirical formula below:(20)$$h=\sqrt{n+m}+\alpha$$
where *n*, *h*, and *m* are the number of nodes in the first, second, and third layer, respectively, and *α* is a random number of [1,10].

According to Equation (20), the range of the number of nodes in the middle layer can be roughly determined as [3,12]. Therefore, the influence of the number of nodes in the interval 3–16 on training error (MSE) was explored. Figure 5 shows the correspondence between the training error size and the number of hidden layer nodes. It can be seen that the training error tends to decrease with the increase in the number of nodes in this range, and the minimum error is obtained when the number of nodes is set to 12. When the number of nodes is greater than 12, the error increases slowly. Therefore, the number of nodes is determined to be 12.

Three parameters, including initial weight, neuron center, and width, determine whether the network can converge to the minimum error and the training speed of the network during the training process. These three parameters will be constantly corrected in the subsequent iterative optimization process and approximate to the values that minimize the global error. The weights, neuron centers, and radial basis widths are initialized to random values between (0, 1).

Based on the above analysis, the RBF neural network structure in this paper is determined to be 3-12-1. In addition, the Gaussian function is used as the excitation function [31]. The initial weight, node center, and radial basis width are initially set as random numbers between (0, 1).

#### 3.1.2. Parameter Setting of PSO Algorithm

Each particle in the PSO algorithm holds the values for its parameters. The position data used in this method are the RBF neural network parameters that need to be tuned. The number of particles *C* = 100 was selected. The dimension of the particle is the dimension of the solution space, which refers to the necessary information contained in the position of the particle, namely, the weight, the neuron center, and the radial basis width, and takes *D* = 60. The maximum number of iterations is *T*_max_ = 200; maximum speed *V*_max_ = 1; learning factor *c*_1_ = *c*_2_ = 1.5; and the inertia weight is updated iteratively according to Equation (6). The termination condition was that the global optimal fitness met the global accuracy requirements, and the MSE index was used as the particle fitness function, that is, the iteration was stopped when there were particles in the particle iteration process that made the global accuracy meet the requirements. In the particle position iteration formula, and are uniformly distributed random numbers between [0, 1].

### 3.2. Comparison of Model Fitting Results

Three prediction models of BP, RBF, and PSO-RBF neural networks are developed for comparison study to demonstrate the superiority of the upgraded GWO-PSO-RBF neural network prediction model established in this study. Figure 6 shows the prediction of the full data set using these four models on the roughness test data, respectively. The unoptimized RBF neural network performs better than the BP neural network fitting results when comparing the benefits and drawbacks of the four different types of neural network prediction effects. The optimized PSO-RBF neural network and GWO-PSO-RBF neural network are better than the ordinary RBF neural network, and the predicted value fits the true value more closely.

The scatter plots and regression lines for the four approaches are shown in Figure 7, where the ordinate represents the predicted values and the abscissa represents the true values. It can be seen that the GWO-PSO-RBF neural network scatter plot is near the fitting line and evenly scattered on both sides. The GWO-PSO-RBF neural network has the best prediction impact thanks to its determination coefficient of 0.919, which is higher than that of the other three models.

With the grinding parameters and the control variable technique, we exhibit the actual as well as the expected roughness in Figure 8. It can be found that when the belt linear speed is low under the condition of high feed speed and grinding depth reduction, large scratches appear on the grinding surface due to uneven grinding, machine vibration, and other factors (Figure 9b). When the belt linear speed and feed speed are high, the lower grinding depth occurs, and the uneven surface pits are produced by grinding due to the existence of factors such as grinding shedding and adhesion (Figure 9a). Due to the existence of these uncertainties, the predicted value learned by the model has a large error from the true value. It is worth noting that the proposed model has a better prediction effect on the whole.

### 3.3. Comparison of Model Evaluation Results

In order to better illustrate the effect of the model, the mean square error (MSE), the root mean square error (RMSE), and the mean absolute percentage error (MAPE) of the prediction results are collected. Each model is simulated four times with the same random data, and the average of the results of the four times is taken as the reference value. The results for each model are summarized in Table 3, Table 4 and Table 5 along with the average values for each index.

According to the comparison of RMSE in Table 4, it can be seen that the evaluation result of the BP neural network model is 0.500, and that of the RBF neural network is 0.245. The result of PSO-RBF is 0.109. The result of GWO-PSO-RBF is 0.082. MSE is the mean square error, and its value is the square of RMSE. The average values of the above four models are 0.252, 0.061, 0.013, and 0.007, respectively. In general, the above three evaluation indexes of the GWO-PSO-RBF hybrid model are the smallest among the four algorithms, which can prove that it improves the accuracy and performance of the grinding roughness prediction to a certain extent, and can effectively predict the grinding roughness. In addition, the RBF neural network based on GWO-PSO optimization is superior to the PSO-RBF neural network in terms of optimization speed and optimization accuracy. Similarly, from Table 5, this index is 46.6% and 23.5%, respectively, after the roughness prediction by a single BP and RBF neural network. The prediction results of the hybrid model optimized by PSO and GWO-PSO decreased to 7.1% and 7.9%, respectively, indicating that the accuracy of the hybrid model combined with the optimization algorithm is higher than that of the single neural network prediction model.

By observing Figure 10, we can see that the single BP model has the worst three accuracy parameters and the lowest prediction accuracy. The prediction accuracy of the RBF neural network model is better than that of the BP neural network model, but the accuracy is still low. In general, whether it is PSO-RBF or GWO-PSO-RBF neural network, the prediction error index of the hybrid model is better than that of the single neural network model, indicating that the accuracy of the hybrid model combined with the optimization algorithm is higher than that of the single neural network prediction model. In addition, the proposed GWO-PSO-RBF neural network is slightly better than the PSO-RBF neural network, maintaining a smaller prediction error.

## 4. Conclusions

In this chapter, a GWO-PSO hybrid optimized RBF neural network is established to establish the mapping relationship between grinding process parameters and the roughness of the titanium alloy abrasive belt to predict surface roughness. The PSO algorithm is used to optimize the parameters of the RBF network to obtain the optimal solution and improve the operation efficiency, and, then, the PSO algorithm is improved by the GWO algorithm to update the iterative formula, which effectively avoids the problem of falling into local optimum due to the decline of convergence ability in the later stage of the algorithm. Then, the structural framework of 3-12-1 is determined by an empirical formula and program debugging, and the main parameter values that need to be used in the algorithm are clarified. Finally, the simulation verifies the accuracy of the algorithm. The simulation results show that the GWO-PSO-RBF improved RBF neural network constructed in this paper can significantly improve the prediction accuracy of the algorithm, and has certain application values.

## Figures and Tables

**Figure 1 materials-16-07224-f001:**
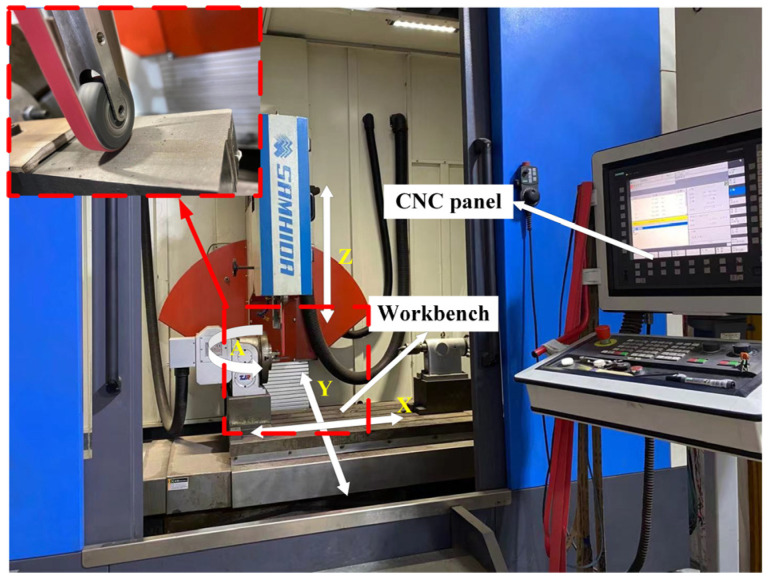
CNC belt grinding machine.

**Figure 2 materials-16-07224-f002:**
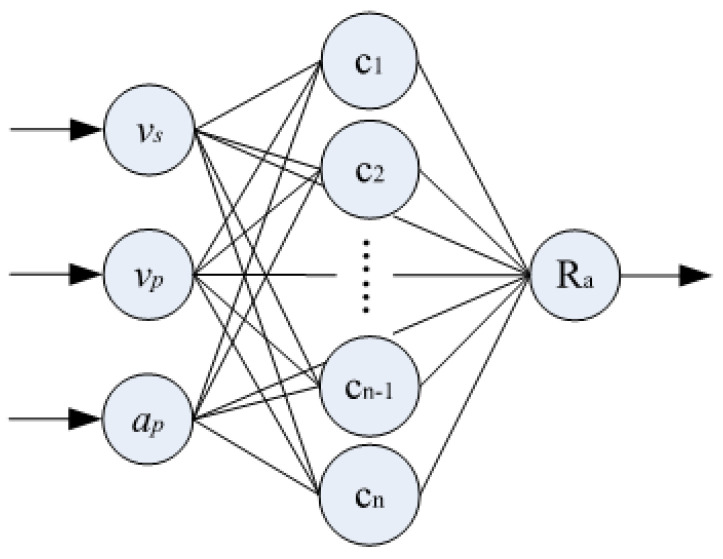
The structure of the RBF neural network [17].

**Figure 3 materials-16-07224-f003:**
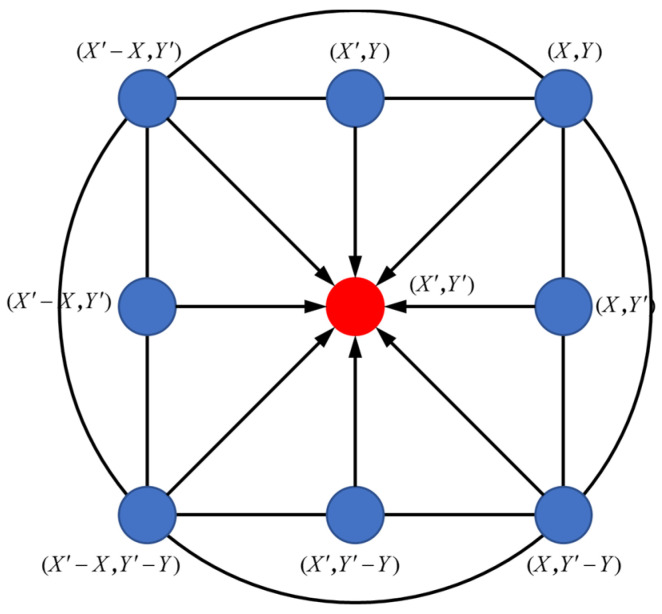
The encirclement strategy of the grey wolf algorithm [27].

**Figure 4 materials-16-07224-f004:**
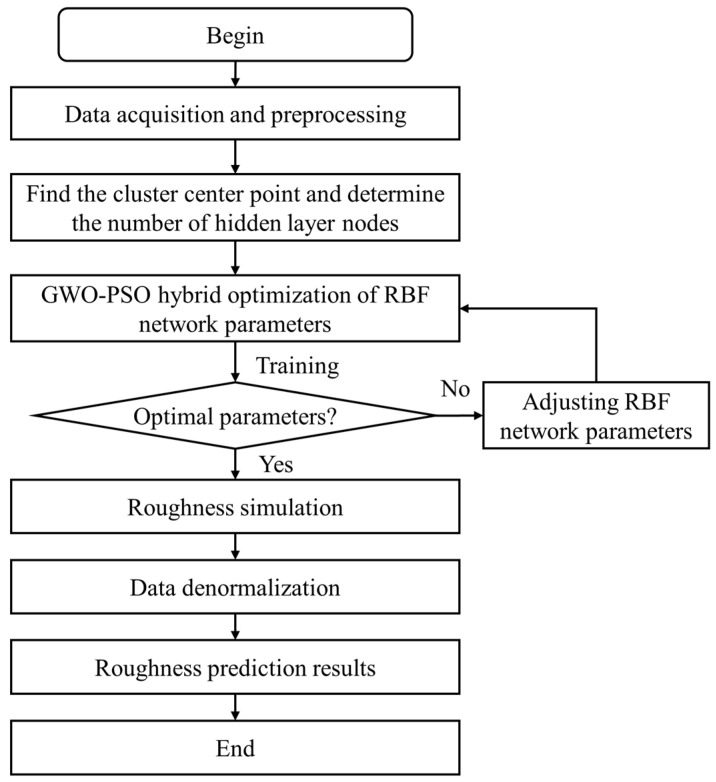
Roughness prediction by GWO-PSO-RBF neural network.

**Figure 5 materials-16-07224-f005:**
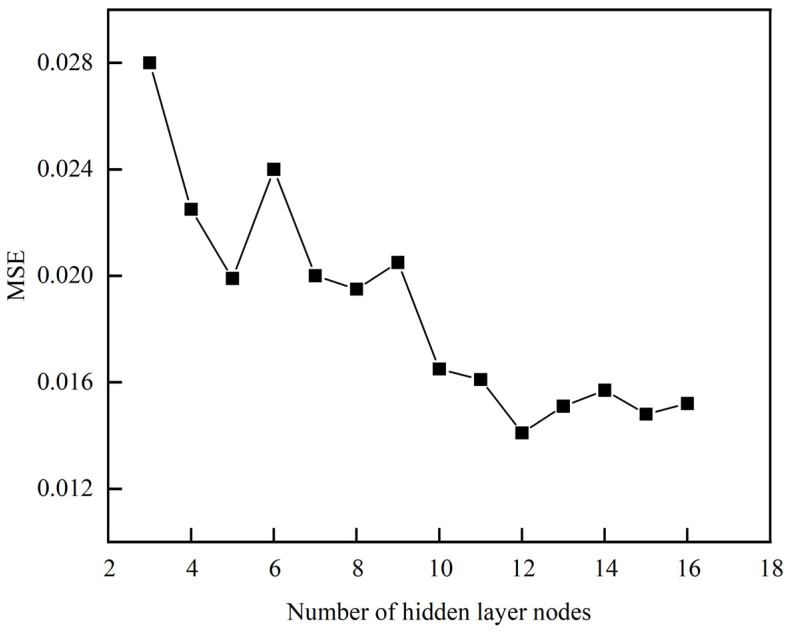
The change of the number of hidden layer nodes.

**Figure 6 materials-16-07224-f006:**
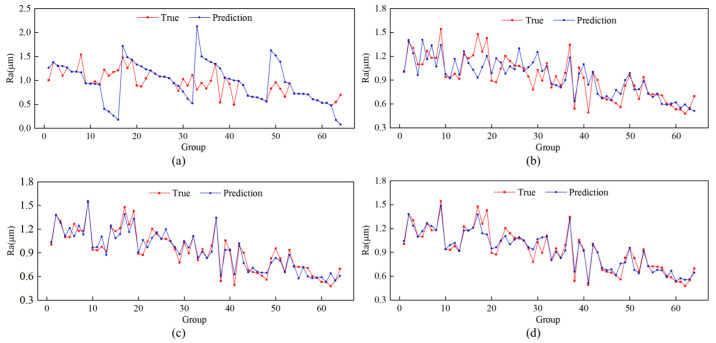
Comparison of prediction results of different models: (**a**) BP neural network; (**b**) RBF neural network; (**c**) PSO-RBF neural network; and (**d**) GWO-PSO-RBF neural network.

**Figure 7 materials-16-07224-f007:**
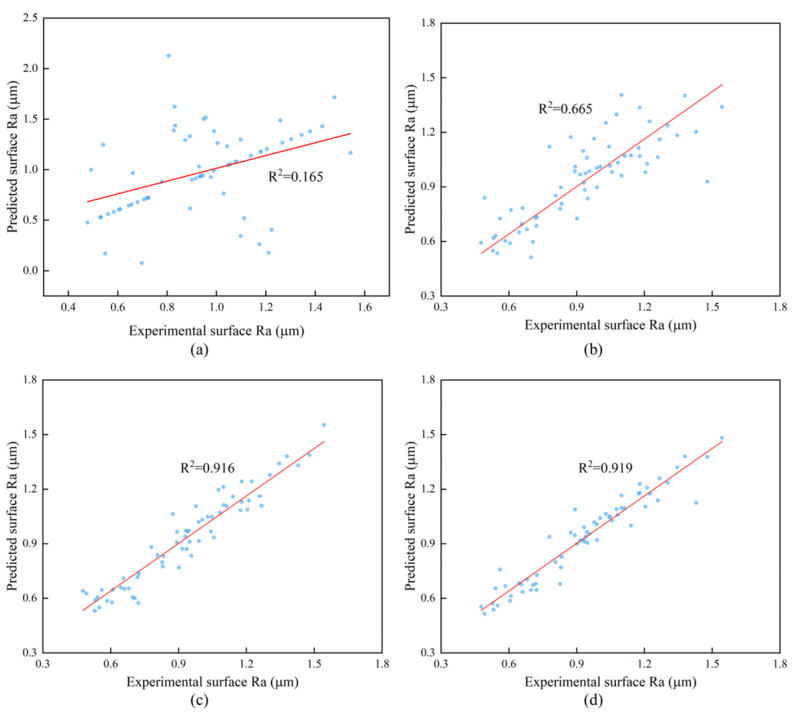
Scatter and regression lines between predicted and real values of the four models: (**a**) BP neural network; (**b**) RBF neural network; (**c**) PSO-RBF neural network; and (**d**) GWO-PSO-RBF neural network.

**Figure 8 materials-16-07224-f008:**
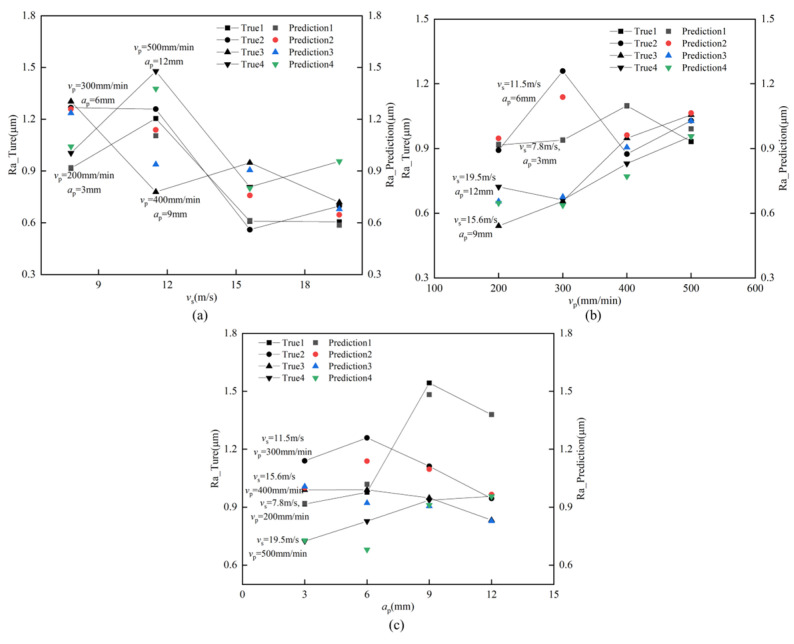
The effect of process parameter variation on roughness prediction: (a) *v*_s_; (**b**) *v*_p_; (**c**) *a*_p_.

**Figure 9 materials-16-07224-f009:**
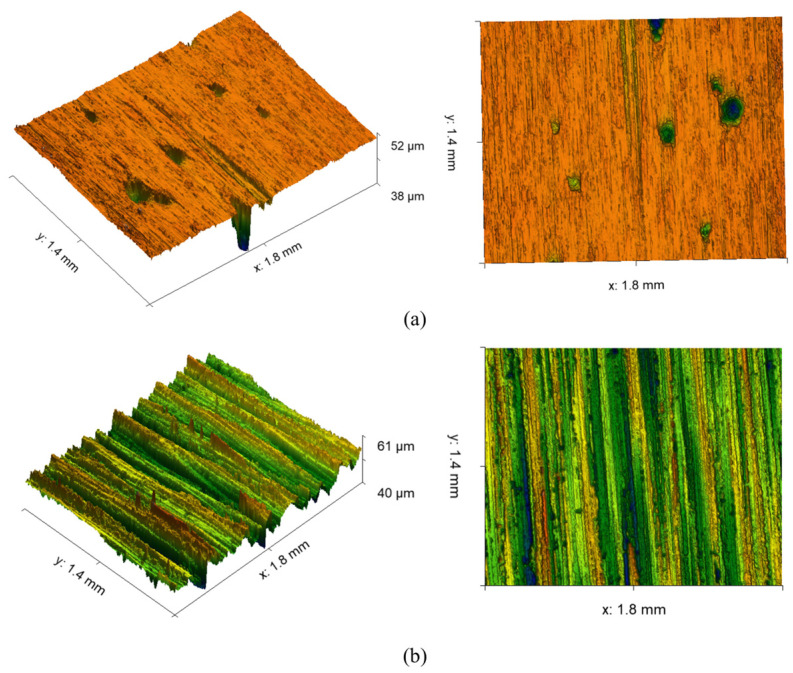
Surface topography of abrasive belt grinding: (**a**) the belt linear speed and feed speed are high during the lower grinding depth; (**b**) the belt linear speed is low under the condition of high feed speed and grinding depth.

**Figure 10 materials-16-07224-f010:**
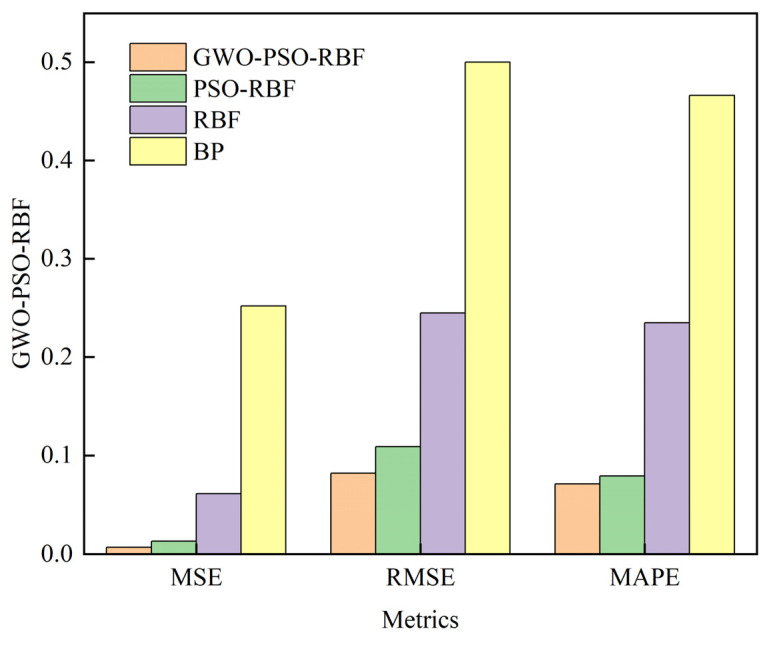
The average value of the three evaluation indicators.

**Table 1 materials-16-07224-t001:** Abrasive belt grinding factor level.

Level	*v*_s_ (m/s)	*v*_p_ (mm/Min)	*a*_p_ (mm)
1	7.8	200	3
2	11.5	300	6
3	15.6	400	9
4	19.5	500	12

**Table 2 materials-16-07224-t002:** Experimental results under different processing parameter conditions (partial result).

*v*_s_ (m/s)	*v*_p_ (mm/Min)	*a*_p_ (mm)	Ra (μm)
7.8	200	3	0.916
7.8	200	6	0.977
7.8	200	9	1.543
7.8	200	12	1.379
11.5	300	3	1.14
11.5	300	6	1.259
11.5	300	9	1.112
11.5	300	12	0.945
15.6	400	3	0.989
15.6	400	6	0.99
15.6	400	9	0.948
15.6	400	12	0.833
19.5	500	3	0.724
19.5	500	6	0.827
19.5	500	9	0.936
19.5	500	12	0.956

**Table 3 materials-16-07224-t003:** Comparison of results for MSE of each model under the test set.

Number	GWO-PSO-RBF	PSO-RBF	RBF	BP
1	0.004	0.007	0.047	0.242
2	0.011	0.008	0.084	0.283
3	0.005	0.024	0.061	0.291
4	0.008	0.013	0.052	0.192
Average value	0.007	0.013	0.061	0.252

**Table 4 materials-16-07224-t004:** Comparison of results for RMSE of each model under the test set.

Number	GWO-PSO-RBF	PSO-RBF	RBF	BP
1	0.064	0.081	0.216	0.492
2	0.105	0.089	0.290	0.532
3	0.071	0.155	0.247	0.539
4	0.089	0.114	0.228	0.438
Average value	0.082	0.109	0.245	0.500

**Table 5 materials-16-07224-t005:** Comparison of results for MAPE of each model under the test set.

Number	GWO-PSO-RBF	PSO-RBF	RBF	BP
1	0.046	0.070	0.208	0.498
2	0.067	0.072	0.214	0.512
3	0.102	0.107	0.321	0.472
4	0.071	0.068	0.198	0.384
Average value	0.071	0.079	0.235	0.466

## Data Availability

Data are contained within the article.

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
