# Peer review of "Surface Roughness Prediction of Titanium Alloy during Abrasive Belt Grinding Based on an Improved Radial Basis Function (RBF) Neural Network"

_materials, 2023, doi:10.3390/ma16227224_

Round 1

Reviewer 1 Report

Comments and Suggestions for Authors

I have reviewed the Paper “Surface roughness prediction of titanium alloy grinding with 2 abrasive belt based on improved RBF neural network”. Following were my observations.

1.                 The paper includes current literature reviews to highlight the ongoing relevance of the chosen topic.

2.                 Table 1 displays the parameters and their respective levels. The approach taken to establish the lower and upper level values for all parameters should be discussed. If any trials were conducted to determine these values, it should be documented in the paper for better clarity.

3.                 The section following Table 1 should specify the Design of Experiments (DOE) used for the experimental trials.

4.                 Table 2 should present a structured DOE table indicating blocks, replicates, etc., with design parameters and the corresponding Ra results.

5.                 There seems to be a discrepancy between the statement in line 102 mentioning 64 experiments and Table 2, which only indicates 16. This inconsistency should be clarified in the paper.

6.                 A reference to Figure 2 needs to be added in the appropriate section.

7.                 Similarly, a reference to Figure 3 should be included where relevant.

8.                 It's essential to explore whether there's a trend or saturation in Mean Squared Error (MSE) when increasing the number of hidden layer nodes beyond 12 in your study. If there is an increasing trend or saturation, a plot illustrating this would be beneficial. If the trend shows a decrease at 12 nodes in the hidden layer, it should be explicitly justified why selecting 14 or 16 nodes may yield a better MSE.

*****************

Comments on the Quality of English Language

MINOR EDITING

Author Response

Dear Editors and Reviewers:

Thank you for your letter and for the reviewers’ comments concerning our manuscript entitled “Surface roughness prediction of titanium alloy grinding with abrasive belt based on improved RBF neural network” (No. materials-2716975). Those comments are all valuable and very helpful for revising and improving our paper, as well as the important guiding significance to our research. We have studied the comments carefully and have made revisions which we hope to meet with approval. Besides, We apologize for the poor language of our manuscript. We worked on the manuscript for a long time and the repeated addition and removal of sentences and sections obviously led to poor readability. We have now worked on both language and readability and have also involved native English speakers for language corrections. We really hope that the flow and language level have been substantially improved.The main corrections are marked in red on the paper and the responds to the reviewer’s comments are as following:

Responds to the reviewer’s comments:

Reviewer #1:

I have reviewed the Paper “Surface roughness prediction of titanium alloy grinding with 2 abrasive belt based on improved RBF neural network”. Following were my observations.

Comment 1): The paper includes current literature reviews to highlight the ongoing relevance of the chosen topic.
Reply: Thank you very much for your comments and suggestions. Firstly, we describe the advantages of abrasive belt grinding in machining titanium alloys. Secondly, the importance of surface roughness prediction for grinding quality improvement is described. Finally, the relevant methods of prediction are summarized, and the significance of this paper is introduced.

Comment 2): Table 1 displays the parameters and their respective levels. The approach taken to establish the lower and upper level values for all parameters should be discussed. If any trials were conducted to determine these values, it should be documented in the paper for better clarity.
Reply: Thank you very much for your comments and suggestions. I am very sorry for the confusion caused to you by such a problem. The selection of experimental parameters is mainly based on the performance of the machine tool and the maximum/minimum parameters based on engineering experience. We use such parameters instead of larger or smaller ones in order to make the prediction closer to our engineering reality. In this regard, we have made supplementary explanations as follows:

The test scheme as given in Table 1 is designed, and a total of 64 grinding experiments are conducted in the pre-experiment. The selection of experimental parameters is mainly based on the performance of the machine tool and the maximum/minimum parameters based on engineering experience. Besides, this testing scheme takes into account the characteristics of the orthogonal test table and neural network (the more training samples, the higher the prediction accuracy). Therefore, we divided the parameters of vs between 7.8m/s–19.5m/s, vp between 200mm/min–500mm/min and ap between 3mm–12mm into four segments at equal in-tervals for orthogonal experiments. After grinding, a portable roughness measurement device was used to determine the surface roughness of the test titanium alloy. During the process, roughness detection is done for five different points, and the final surface roughness value is calculated by taking the arithmetic average of the five measurements.

Comment 3): The section following Table 1 should specify the Design of Experiments (DOE) used for the experimental trials.
Reply: Thank you very much for your comments and suggestions. For the data in Table 1, we have made supplementary explanations and more detailed explanations, which are as follows:

The test scheme as given in Table 1 is designed, and a total of 64 grinding experiments are conducted in the pre-experiment. The selection of experimental parameters is mainly based on the performance of the machine tool and the maximum/minimum parameters based on engineering experience. Besides, this testing scheme takes into account the characteristics of the orthogonal test table and neural network (the more training samples, the higher the prediction accuracy). Therefore, we divided the parameters of vs between 7.8m/s–19.5m/s, vp between 200mm/min–500mm/min and ap between 3mm–12mm into four segments at equal in-tervals for orthogonal experiments. After grinding, a portable roughness measurement device was used to determine the surface roughness of the test titanium alloy. During the process, roughness detection is done for five different points, and the final surface roughness value is calculated by taking the arithmetic average of the five measurements.

Comment 4): Table 2 should present a structured DOE table indicating blocks, replicates, etc., with design parameters and the corresponding Ra results.
Reply: Thank you very much for your comments and suggestions. In Table 2, we intended to display some data in the format of data sets, but this meaning was not well expressed in the original description, so the relevant contents were improved as follows:

As a result, the dataset of grinding parameters and corresponding roughness was built. Table 2 displays some data of the dataset.

Table 2. Experimental results under different processing parameter conditions (Partial result).

vs (m/s)

vp (mm/min)

ap (mm)

Ra (μm)

7.8

200

3

0.916

7.8

200

6

0.977

7.8

200

9

1.543

7.8

200

12

1.379

11.5

300

3

1.14

11.5

300

6

1.259

11.5

300

9

1.112

11.5

300

12

0.945

15.6

400

3

0.989

15.6

400

6

0.99

15.6

400

9

0.948

15.6

400

12

0.833

19.5

500

3

0.724

19.5

500

6

0.827

19.5

500

9

0.936

19.5

500

12

0.956

Comment 5): There seems to be a discrepancy between the statement in line 102 mentioning 64 experiments and Table 2, which only indicates 16. This inconsistency should be clarified in the paper.
Reply: Thank you very much for your comments and suggestions. I am very sorry for the data because the data is confidential. But in good persuasion, we present some of the experiment results as displayed in Table 2 to increase the authenticity of this study. In this response, we provide additional data for review by reviewers, but we regret that we are unable to provide all data.

vs

vp

ap

Ra

7.8

500

3

0.932

7.8

500

6

1.211

7.8

500

9

1.179

7.8

500

12

1.004

7.8

400

3

1.098

7.8

400

6

1.174

7.8

400

9

1.303

7.8

400

12

1.223

7.8

300

3

0.939

7.8

300

6

1.267

7.8

300

9

1.179

7.8

300

12

1.098

7.8

200

3

0.916

7.8

200

6

0.977

7.8

200

9

1.543

7.8

200

12

1.379

11.5

500

3

1.076

11.5

500

6

1.029

11.5

500

9

1.043

11.5

500

12

1.478

11.5

400

3

1.429

11.5

400

6

0.874

11.5

400

9

0.779

11.5

400

12

1.082

11.5

300

3

1.14

11.5

300

6

1.259

11.5

300

9

1.112

11.5

300

12

0.945

11.5

200

3

1.204

11.5

200

6

0.892

11.5

200

9

0.893

11.5

200

12

1.047

Comment 6): A reference to Figure 2 needs to be added in the appropriate section.
Reply: Thank you very much for your comments and suggestions. Based on the basic RBF network structure, we redraw according to the input and output of this study, and the corresponding references are cited in the diagram notes. The revise as follows:

Figure 2. The structure of RBF neural network [17].

Comment 7): Similarly, a reference to Figure 3 should be included where relevant.
Reply: Thank you very much for your comments and suggestions. The corresponding references are reflected in the figure notes. Thanks again for the tips of the review experts.

Figure 3. The encirclement strategy of the grey Wolf Algorithm [27].

Comment 8): It's essential to explore whether there's a trend or saturation in Mean Squared Error (MSE) when increasing the number of hidden layer nodes beyond 12 in your study. If there is an increasing trend or saturation, a plot illustrating this would be beneficial. If the trend shows a decrease at 12 nodes in the hidden layer, it should be explicitly justified why selecting 14 or 16 nodes may yield a better MSE.
Reply: Thank you very much for your comments and suggestions. We are very sorry for the omission here. In the original experiment, we tried the influence of nodes greater than 12 on the accuracy, but we did not reflect it in the figure, which made the relevant content not rigorous enough. In the revised figure, relevant information has been supplemented, and relevant modifications are as follows:

According to Equation (21), the range of the number of nodes in the middle layer can be roughly determined as [3,12]. Therefore, the influence of the number of nodes in the interval 3-16 on training error (MSE) was explored. Figure 5 shows the correspondence between the training error size and the number of hidden layer nodes. It can be seen that the training error tends to decrease with the increase in the number of nodes in this range, and the minimum error is obtained when the number of nodes is set to 12. When the number of nodes is greater than 12, the error increases slowly. Therefore, the number of nodes is determined to be 12.

Figure 5. The change of the number of hidden layer nodes.

Reviewer 2 Report

Comments and Suggestions for Authors

Authors have worked on developing a model that can predict surface roughness. The work looks interesting and gives new insight into surface roughness study. Some comments as below:

1. Not clear how the algorithm will be used in real-time modification of machining process parameters.

2. Definitely the PSO algorithm can give optimised process parameters but no clarity on how the process will be monitored and controlled in real-time.

3. No proper justification on why there is significant improvement the prediction accuracy.

Comments on the Quality of English Language

The paper requires minor English revision, particularly in reframing some sentences.

Author Response

Dear Editors and Reviewers:

Thank you for your letter and for the reviewers’ comments concerning our manuscript entitled “Surface roughness prediction of titanium alloy grinding with abrasive belt based on improved RBF neural network” (No. materials-2716975). Those comments are all valuable and very helpful for revising and improving our paper, as well as the important guiding significance to our research. We have studied the comments carefully and have made revisions which we hope to meet with approval. Besides, We apologize for the poor language of our manuscript. We worked on the manuscript for a long time and the repeated addition and removal of sentences and sections obviously led to poor readability. We have now worked on both language and readability and have also involved native English speakers for language corrections. We really hope that the flow and language level have been substantially improved.The main corrections are marked in red on the paper and the responds to the reviewer’s comments are as following:

Responds to the reviewer’s comments:

Reviewer #2:

Authors have worked on developing a model that can predict surface roughness. The work looks interesting and gives new insight into surface roughness study. Some comments as below:

Comment 1): Not clear how the algorithm will be used in real-time modification of machining process parameters.
Reply: Thank you very much for your comments and suggestions. Through the prediction of surface roughness, we establish a mapping model of process parameters to surface quality (roughness in this paper). The further work is to establish the corresponding parameter optimization method through the prediction model to optimize the process parameters in reverse, that is, the mapping of machining quality to process parameters. Finally, through the real-time monitoring of hardware equipment and the real-time optimization of process parameters through the algorithm model, the optimization of machining quality can be realized.

Comment 2): Definitely the PSO algorithm can give optimised process parameters but no clarity on how the process will be monitored and controlled in real-time.
Reply: Thank you very much for your comments and suggestions. The GGO-PSO optimization method proposed in this paper mainly optimizes the RBF optimization process. After the optimized model is trained, the trained model is used for prediction and subsequent parameter optimization. We use the trained model for prediction, and do not optimize the model in real time.

Comment 3):
No proper justification on why there is significant improvement the prediction accuracy.
Reply: Thank you very much for your comments and suggestions. In 2.3.2, we added the reasons why PSO was included in RBF and explained the mechanism of the improved accuracy. In addition, the first paragraph of 2.3.3 also explains the impact of GWO on particle swarm optimization, and the relevant contents are as follows:

2.3.2à Three parameters in RBF neural network need to be set artificially: weight, node center, and radial basis width. The traditional RBF neural network often uses gradient descent iterative optimization, but this method not only has a poor training effect, a long time but also is easy to fall into local optimum so that the global effect cannot reach the best. The parameter selection of the RBF neural network is essentially an optimization process, so the PSO optimization algorithm is used to optimize the parameters that need to be set manually in RBF neural network and select the optimal values.

2.3.3à The particle swarm is easy to aggregate in the final iteration, which reduces its searchability and causes the roughness prediction result to fall into the local optimum. Additionally, depending solely on the global optimum to direct parameter iteration will cause the subsequent iterations to move more slowly and reduce the capacity for convergence. The grey wolf method [27] will be incorporated into this study to enhance the parameter iteration formula and address the issues.

Round 2

Reviewer 1 Report

Comments and Suggestions for Authors

Paper is improved by incorporating all suggestions/ comments.